

**Trend Quality Ozone from NPP OMPS: the Version 2 Processing**
Richard McPeters[1], Stacey Frith[2], Natalya Kramarova[1], Jerry Ziemke[3], and Gordon Labow[2]

**Abstract.** A version 2 processing of data from two ozone monitoring instruments on Suomi
NPP, the OMPS nadir ozone mapper and the OMPS nadir ozone profiler, has now been
completed. The previously released data were useful for many purposes but were not suitable for
use in ozone trend analysis. In this processing, instrument artifacts have been identified and
corrected, an improved scattered light correction and wavelength registration have been applied,
and soft calibration techniques were implemented to produce a calibration consistent with data
from the series of SBUV/2 instruments. The result is a high quality ozone time series suitable for
trend analysis. Total column ozone data from the OMPS nadir mapper now agree with data from
the SBUV/2 instrument on NOAA 19 with a zonal average bias of -0.2% over the 60°S to 60°N
latitude zone. Differences are somewhat larger between OMPS nadir profiler and N19 total
column ozone, with an average difference of -1.1 % over the 60°S to 60°N latitude zone and a
residual seasonal variation of about 2% at latitudes higher than about 50 degrees. For the profile
retrieval, zonal average ozone in the upper stratosphere (between 2.5 and 4 hPa) agrees with that
from NOAA 19 within ±3% and an average bias of -1.1%. In the lower stratosphere (between 25
and 40 hPa) agreement is within ±3% with an average bias of +1.1%. Tropospheric ozone
produced by subtracting stratospheric ozone measured by the OMPS limb profiler from total
column ozone measured by the nadir mapper is consistent with tropospheric ozone produced by
subtracting stratospheric ozone from MLS from total ozone from the OMI instrument on Aura.
The agreement of tropospheric ozone is within 10% in most locations.

[1]NASA Goddard Space Flight Center, Greenbelt Maryland
[2]Science Systems and Applications Inc., Lanham, Maryland
[3]Morgan State University, Baltimore. Maryland



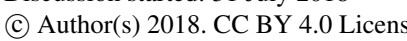

## 1. Introduction

NASA has been measuring ozone from space since the launch of the Backscatter
Ultraviolet (BUV) instrument on Nimbus 4 in 1970. The series of follow-on instruments, SBUV
(Solar Backscatter Ultraviolet) and TOMS (Total Ozone Mapping Spectrometer) on Nimbus 7
and SBUV/2 instruments on NOAA 9, 11, 14, 16, 17, 18, and 19 produced a long term time
series of global ozone observations. Under NASA's MEaSUREs (Making Earth System data
records for Use in Research Environments) program, data from this series of instruments were
re-processed to create a coherent ozone time series. Inter-instrument comparisons during periods
of overlap as well as comparisons with data from other satellite and ground based instruments
were used to evaluate the consistency of the record and make careful calibration adjustments as
needed (McPeters et al., 2013). The result is an ozone data record suitable for trend studies that
we designated the Merged Ozone Data (MOD) time series (Frith et al., 2014). Ozone instruments
on the Suomi-NPP spacecraft and the planned series of JPSS (Joint Polar Satellite System)
spacecraft will now be used to continue this series of measurements in order to document long-
term ozone change.
The Suomi National Polar-orbiting Partnership (Suomi NPP) is a joint NOAA/NASA
mission that collects and distributes remotely sensed land, ocean, and atmospheric data to the
meteorological and global climate change communities. Suomi NPP was launched October 28,
2011. The Ozone Mapper Profiler Suite (OMPS) on NPP consists of three instruments - the
ozone total column Nadir Mapper (NM), an instrument similar to the TOMS and OMI ozone
mapping instruments, the Nadir Profiler (NP), an instrument similar to the SBUV and SBUV/2
profilers, and the Limb Profiler (LP), an instrument that measures the ozone vertical distribution
using light scattered from the Earth's limb. Details of the OMPS instruments and mission are
given by Flynn at al. (2006).
The purpose of the version 2 processing of data from the two OMPS nadir sensors, which is
the subject of this paper, is to correct various instrument artifacts and to apply an updated
calibration that will be consistent with data from earlier instruments. Only the reprocessed
version 2 data from the two nadir instruments will be discussed here. While some comparisons
with data from the Limb Profiler will be shown in this paper, detailed LP validation results will
be discussed in other papers.
## 2. The OMPS Nadir Mapper and Nadir Profiler

The OMPS nadir mapper (NM) is a nadir viewing, wide swath, ultraviolet-visible imaging
spectrometer that provides daily global measurements of the solar radiation backscattered by the
Earth's atmosphere and surface, along with measurements of the solar irradiance. It shares a
telescope with the OMPS nadir profiler (NP) spectrometer. A dichroic filter splits light from the
telescope into two streams. Most of the 310-380 nm light is transmitted to the NM instrument,
while most of the 250-300 nm light is reflected to the NP instrument. The transition between
reflection and transmission occurs between 300 and 310 nm, the wavelength overlap region. The





detector for each instrument is a 340x740 pixel CCD (Charge Coupled Device). For more details
on the instruments and sensors see Seftor et al. (2013).
Unlike the heritage TOMS instruments which measured ozone using a photomultiplier
detector at six discrete wavelengths (from 306 to 380 nm, depending on the instrument), the NM
instrument measures the complete spectrum from 300 to 380 nm at an average spectral resolution
of 1.1 nm. The OMPS-NM sensor has a 110 degree cross-track field of view, with 35 discrete
cross-track bins. The 0.27 µm along track slit width produces a 50 km spatial resolution near
nadir. An algorithm uses the radiance and irradiance measurements to infer total column ozone.
As illustrated in Figure 1, the OMPS NM makes 400 individual scans per orbit with 35 across-
track measurements in each scan, which provides full global coverage of the sunlit Earth every
day.
The OMPS nadir profiler (NP) has a 16.6 µm cross-track slit and a 0.26 µm along-track slit
width, producing a ground FOV cell size of 250 km by 250 km when exposed for a 38 second
sample time. The OMPS NP instrument makes 80 measurements per orbit, resulting in full
global coverage approximately every 6 days. The NP measures the complete spectrum from 250
to 310 nm with a 1.1 nm bandpass. Because the NP itself only makes measurements up to a
maximum wavelength of 310 nm, the longer wavelengths that are needed in the retrievals at high
latitudes must be taken by averaging the overlap cells from the NM instrument, the 5 central
cross track cells in 5 along track scans.
**3. The Version 2 Processing**
The goal of the version 2 processing is to produce ozone data sufficiently accurate to be
used to continue the Merged Ozone Data (MOD) time series. This time series is a unified multi-
instrument ozone data set created by merging data from a series of SBUV and SBUV/2
instruments beginning with the original BUV instrument launched on Nimbus 4 in 1970 and
extending to the SBUV/2 instrument on NOAA 19, which continues to operate. Data from these
instruments were recently reprocessed as version 8.6 with a consistent calibration to create a
coherent ozone time series (McPeters et al., 2013). The MOD data set created from this series is
described in detail by Frith et al. (2014). Figure 2 shows the MOD fit to data from three recent
SBUV/2 instruments, on NOAA 16, 18, and 19, for which good data are available during the
OMPS observation period. Comparison with ozone from ground networks shows that total ozone
in the MOD series is consistent to within about a percent for the recent data. Data from the
OMPS NP and NM instruments will be used to extend this MOD data record.
In the version 2 processing we use the latest version of the Level 1 data, the dataset of
calibrated radiance measurements from NM and NP that implements a refined calibration for
both instruments (Seftor et al., 2014) and corrects for several instrument effects. Both the NM
and NP L1b data now use an improved set of calibration coefficients that exhibit smoother
wavelength-to-wavelength behavior and provide a wavelength registration that accounts for
intra-orbital (for the NM) and intra-seasonal (for the NP) shifts that were identified in analysis of
the data. A small bandpass error in the NP instrument near 295 nm was corrected, and errors in
the pre-launch calibration measurements in the dichroic transition region (300 - 310 nm) for both



instruments were identified and corrected. The daily dark current correction has been refined for
each instrument.
Soft (in orbit) calibration techniques were used to refine the instrument calibration. The
NM pre-launch calibration of the 331 nm channel, which is used to determine reflectivity, was
not adjusted at nadir since the measured radiance over ice matched the expected radiance
(determined from other instruments such as Earth Probe TOMS and OMI) to within 1%. Cross-
track adjustments to this channel to "flatten" the 331 nm reflectivity calculation over ice were
then determined and applied. Similarly, the nadir radiance at 317 nm, which is the channel used
to determine ozone, was not changed; the off-nadir radiances were then adjusted to take out any
cross track ozone dependence. The 317 and 331 nm NM nadir radiances are also used in the NP
algorithm retrieval, with no adjustments applied. For the NM radiances at 312 nm, which are
used in the NP algorithm but not in the NM algorithm, an adjustment was determined, and
applied to minimize the final retrieval residuals. Similarly, the NP 306 nm radiances were
adjusted to minimize the final residuals. The calibrations were not explicitly adjusted to agree
with the NOAA 19 SBUV/2 calibration, so NOAA 19 comparisons can be used for validation.
The algorithm used to retrieve total column ozone from the NM is very similar to the v8.5
algorithm used in the processing of data from Aura OMI instrument as described by Bhartia
(2007), and Bhartia et al. (2004). The basic algorithm uses two wavelengths to derive total
column ozone, one wavelength with weak ozone absorption (331 nm) to characterize the
underlying surface and clouds, and the other at a wavelength with strong ozone absorption (317
nm). The ozone retrieval algorithms for both the NP and NM instruments now use the Brion/
Daumont / Malicet ozone cross sections (Brion et al., 1993) to be consistent with other data sets
in the MOD time series.
The NP retrieval algorithm uses 12 discrete wavelengths to retrieve ozone profiles
employing Rodgers' optimal estimation technique (Bhartia et al., 2013). It is very similar to the
v8.6 algorithm used to reprocess the SBUV and SBUV/2 data sets (McPeters et al., 2013) used in
the MOD time series. While the vertical resolution of an OMPS NP ozone retrieval is somewhat
coarse in comparison with the LP sensor, about 8 km resolution in the stratosphere, NP provides
valuable data for the continuation of the historical SBUV/2 ozone data record, and for validation
of the OMPS LP retrievals.
**4. Total Column Ozone Comparisons**
The accuracy and stability of the OMPS ozone data record has been evaluated through
comparisons with ground-based observations and comparisons with other satellite data sets. The
worldwide network of Dobson and Brewer stations has been used for years for ground-based
validation of total column ozone. For satellite validation of total ozone, comparisons with the
MOD data set are used as a primary standard for this evaluation. Validation of profile ozone (in
section 5) will use data from balloon sondes, data from the currently operating SBUV/2
instrument on NOAA 19, and data from the microwave limb sounder (MLS) on the Aura
spacecraft.



Figure 3 compares average ozone from 52 ground based Brewer and Dobson stations in the
northern hemisphere with coincident observations of ozone measured by the NM instrument over
the individual stations (Labow et al., 2013). Northern hemisphere comparisons are shown
because the network density is much better in the northern hemisphere than in the southern, and
comparisons in a single hemisphere will illuminate any seasonally dependent errors. Such
comparisons have been shown capable of detecting instrument changes over the long term of a
few tenths of a percent (McPeters at al., 2008). The comparison covers the period from April
2012 through the end of 2016. Figure 3 shows that the agreement of NM total ozone is mostly
within half a percent. The linear fit in Figure 3 shows that OMPS NM has almost no drift in
ozone relative to the ground observations and an average bias of less than 0.2%.
The comparison of ozone from the NM instrument with ozone from the MOD (merged
ozone dataset) time series shown in Figure 4 illustrates the improved accuracy of the version 2
processing. The monthly zonal average ozone, area weighted for the latitude zone from 60°S to
60°N, is plotted. Because ozone is derived from measurements of backscattered sunlight, data are
not always available in winter months at latitudes above 60 degrees. MOD ozone for this time
period is based on combining ozone from SBUV/2 instruments on three satellites, NOAA 16, 18,
and 19. For the period from March 2014 to 2017 only the instrument on NOAA 19 was
operational. The lower panel in Figure 4 shows the NM monthly average ozone for the old
version 1 processing (dashed red curve) and the new version 2 processing (solid blue curve)
along with MOD average ozone (orange curve). The upper panel shows the percent difference of
version 1 and version 2 ozone from MOD ozone. Where in version 1 NM ozone was on average
1% high relative to MOD, in the version 2 processing it is 0.2% low. There is a small relative
trend between NM and MOD of 0.8% per decade. This relative trend could be due to either NM
or to an aging NOAA 19 SBUV/2 instrument. Further comparisons will be needed to distinguish
between the two possibilities.
Figure 5 is the same plot but for total column ozone measured by the NP instrument. NP
total column ozone is derived by integrating the retrieved ozone profiles. In principle, this should
be more accurate over a broad range of solar zenith angles than ozone derived from the limited
wavelength range of the NM instrument. Here the average relative bias of about +1.4% in
version 1 is reduced to -1.05% in version 2. This bias disagreement between NM and NP means
that there is a small inconsistency between the two instruments that has not been resolved. As
noted earlier, NP only measures wavelengths up to 310 nm, so the longer wavelengths used in
the retrieval are taken from the NM instrument. This means the NP total column ozone is
influenced by the relative calibration of the NM instrument and that the calibrations are not
completely consistent at the one percent level. This issue of the relative calibration inconsistency
is being studied. There is a relative drift of NP ozone relative to MOD that is similar to that for
the NM instrument, of 0.5% per decade. To the extent that the NP and NM instruments have
independent calibrations, this suggests that the small relative drift is due to the NOAA 19
SBUV/2 instrument calibration.
Figure 6 shows the latitude dependence relative to MOD of the version 2 ozone from the
mapper and from the profiler. The lower panel plots ozone averaged for five Marches from 2013
through 2016, while the upper panel shows the percent difference from MOD for the same





months. The latitude dependence of ozone varies greatly by season so it is useful to examine
individual months, and latitude coverage is maximum near an equinox. The NM instrument has
very little latitude dependence except at the highest southern latitudes where ozone is low. The
NP instrument has the bias as noted in Figure 5 and likewise has little latitude dependence at low
to mid latitudes. The higher ozone (by 2 to 3 percent) for retrievals at latitudes greater than 50°
may reflect an inconsistency between the longer wavelengths used in the profile retrieval that
come from NM and the shorter wavelengths (<305 nm) that come from the NP itself. A zenith
angle dependence will lead to a seasonal variation for the NP high latitude ozone. This will be
discussed in the profile comparison section.
**5. Ozone Profile Comparisons**
The long-term behavior of ozone as a function of altitude is in some ways more interesting
than the behavior of total column ozone because it can be used to confirm the accuracy of
various model predictions. However, the accuracy of these measurements is more difficult to
validate (Hassler et al., 2014). Data from the ozone sonde network can be used to validate the
profile in the troposphere and lower stratosphere, while satellite data can be used to validate the
middle to upper stratospheric results. There are ground-based measurements of the ozone vertical
distribution by LIDAR and by microwave sounders, but such measurements are very sparse.
There are umkehr measurements by Dobson and Brewer instruments, but vertical resolution is
coarse and uncertainty is high, especially when aerosols are present.
Looking at ground based comparisons of ozone in the lower stratosphere first, Figure 7
compares NP ozone profiles with ozone measured by ECC ozone sondes from one station, Hilo,
Hawaii, a subtropical station with a good time series of sonde launches. The sonde data are from
the SHADOZ network under which the sonde data were reprocessed to apply the most recent
corrections (Witte et al., 2016). For this figure, all 33 of the sondes launched in 2016 were
averaged. The coincident profiles measured by NP were usually within one degree of latitude
and within fifteen degrees of longitude. The comparison shows that in the lower stratosphere NP
agrees with sonde data to within ±5%. Only altitudes between 10 and 50 hPa (approximately 20
to 32 km) are shown because the SBUV nadir ozone retrieval algorithm produces little profile
information on the distribution of ozone below 20 km. But it should be noted that the column
amount of ozone in the troposphere is retrieved accurately (Bhartia et al., 2013), as evidenced by
the fact that total column ozone from an SBUV retrieval is accurate to one percent or better
(McPeters et al., 2013). This accuracy is critical to the derivation of tropospheric ozone
discussed in section 6.
For the middle to upper stratosphere, monthly zonal means comparisons with other satellite
observations of the ozone vertical distribution is the best approach for evaluating the accuracy of
the version 2 NP results. Figure 8 shows the time dependent difference of NP from the NOAA 19
SBUV/2 retrievals averaged over low to middle latitudes (40°S to 40°N), for the upper
stratosphere (2.5 - 4 hPa), lower stratosphere (25 - 40 hPa), and total column ozone. Comparing
with N19 only rather than MOD gives a bit more uniformity for the time dependent profile
comparison. In both the upper stratosphere and lower stratosphere the vsn 2 ozone agrees with



the N19 ozone to within about one percent, where in the NP version 1 retrievals, ozone was
higher by 4% and 6% respectively. There is no evidence of a significant time dependent
difference. There is the bias in total column ozone as noted earlier of a bit over one percent that
is likely produced when wavelengths from the NP instrument are combined with wavelengths
from the NM instrument in the 300 to 310 nm overlap region. Since the NM bias was near zero,
this inconsistency of the NM and NP total column ozone remains to be addressed.
Ozone agreement as a function of altitude is shown in Figure 9 where ozone in low to
middle latitudes is averaged for five Junes from 2012 through 2016. Selecting a single month for
this comparison allows us to see any seasonal effect that might be suppressed in the annual
average. As will be shown later, there are seasonal variations in NP ozone at high latitudes. The
stratospheric ozone mixing ratio is plotted for OMPS NP vsn 2, for NOAA 19 SBUV/2, for the
Aura Microwave Limb Sounder (MLS) (Froidevaux et al., 2008), and for the OMPS limb
profiler (LP). The right panel shows the agreement of the OMPS NP vsn 2 ozone profile with
each of the three other profile measurements by plotting the percent difference from each.
Agreement is almost always within ±5%, which experience has shown to be fairly good
agreement for profile comparisons. While agreement in the upper stratosphere and lower
stratosphere shown in Figure 8 was good, Figure 9 shows that there is a significant underestimate
of ozone relative to NOAA 19, MLS and LP in the 6 to 10 hPa region. This is partly the source
of the disagreement in total column ozone. It has been noted in other comparisons (*Hassler et al.,*
*2014*), that NOAA 19 ozone is a bit high in the upper stratosphere relative to MLS profiles, and a
similar result is seen here for the NP retrievals.
The NP vsn 2 ozone has somewhat different behavior at low to mid latitudes than at high
latitudes. The ozone anomaly, the percent difference of NP ozone from the NOAA 19 SBUV
ozone, is shown for low to mid latitudes (<45°) in Figure 10, and for higher latitudes (>45°) in
Figure 11. For each figure the anomaly is shown for total column ozone (lower panel), for lower
stratospheric ozone (layer from 25 hPa to 40 hPa) in the middle panel, and for upper
stratospheric ozone (layer from 2.5 hPa to 4 hPa) in the upper panel. Figure 10 shows that vsn 2
ozone at latitudes below 45° agrees well with N19 ozone, while Figure 11 shows that at latitudes
at 50° and above ozone has a significant seasonal dependence that differs from that of N19 with
about 2 to 4% amplitude. This difference is likely another manifestation of the inconsistency that
comes from using NM wavelengths in the NP retrievals. This calibration inconsistency is small
but we are working to resolve it in order to produce a better NP ozone product.
**6. Tropospheric Ozone from OMPS**
Ziemke et al. (2011, 2014, and references therein) have shown that tropospheric ozone can
be derived by subtracting stratospheric ozone from total column ozone. This technique has most
recently been applied by subtracting stratospheric ozone measured by the Aura MLS instrument
from total column ozone measured by the Aura OMI instrument. The OMI/MLS tropospheric
ozone data time series currently spans over 12 years and has been a central data product for each
of the BAMS State of the Climate Reports since 2013 and will be used in the upcoming
international Tropospheric Ozone Assessment Report.





The OMPS ozone measurements can also be used to calculate tropospheric ozone and
continue the current OMI/MLS time series of measurements should either of the Aura
instruments fail. Because the OMPS instrument suite includes both a total ozone mapper (NM)
and a limb profiler (LP), a similar technique can be applied as with OMI/MLS. Figure 12 shows
the tropospheric ozone time series for two locations in the tropics, Java and Brazil, and two
locations at northern mid-latitudes, Beijing, and Washington DC. In each case the red dashed
curve shows tropospheric ozone derived by subtracting MLS stratospheric ozone from OMI total
column ozone. For comparison, the blue solid curve shows the same tropospheric ozone derived
by subtracting stratospheric ozone from the OMPS LP from total column ozone from the NM.
While there are some small differences the overall agreement is quite good. Data on tropospheric
ozone from the NP plus LP combination can be used to continue the tropospheric ozone time
series.

**7. Data Availability**

NPP OMPS version 2 data will shortly be available online from the Goddard DISC:
https://disc.gsfc.nasa.gov. Data for the NM mapper and the NP profiler are currently being
converted to HDF5 format for inclusion in the DISC data archive. The OMPS NM ozone data
are also available in ascii form from our online site: https://acd-ext.gsfc.nasa.gov/anonftp/toms/
in the subdirectory omps_tc. Data from the NOAA 19 SBUV/2 can also be found here under
subdirectory sbuv. The v8.6 MOD data used as our standard for comparison are available from:
https://acdb-ext.gsfc.nasa.gov, then click on "Data_services" and then on "Merged ozone data".

**8. Conclusions**

The OMPS nadir mapper (NM) has proven to be a very stable instrument. Comparison with
a network of 52 Northern Hemisphere ground based Dobson and Brewer instruments shows very
good agreement over the four years of operation, agreeing within ±0.5% with near zero trend.
Total column ozone from the OMPS nadir mapper agrees with MOD ozone and with NOAA 19
SBUV/2 ozone with a bias or -0.2% and a small time-dependent drift of 0.8% per decade. It is
possible that this time dependence could be due to the aging NOAA 19 instrument and its
drifting orbit.
The nadir profiler (NP) has likewise been very stable. NP total column ozone has a time
dependence of only 0.5% per decade relative to MOD or NOAA 19. The bias of -1.1% (60°S -
60°N) is small but inconsistent with ozone from NM. The calibration of the NP instrument near
300 nm is being examined to understand this inconsistency. NP ozone in the upper stratosphere
(2.5 - 4 hPa) and in the lower stratosphere (25 - 40 hPa) agrees well with ozone from NOAA 19
profiler, with an average difference of -1.1% and +1.1% respectively at latitudes below 50°. The
retrievals for higher latitudes exhibit a strong seasonal variation of about ±2%, both in layer
ozone and in total column ozone.
Ozone data from these instruments can now be considered "trend quality," usable to extend
the data record from previous instruments to create an accurate time series. Data from NP at





latitudes above 50° appears to be stable but must be used with a bit of caution because of its
residual seasonal variation and because the bias, while small, can be different than at lower
latitudes.

**Acknowledgments**
The OMPS nadir profiler and nadir mapper were built by Ball Brothers for flight on the joint
NASA / NOAA NPP satellite. We thank the many people who have worked over the years to
understand the behavior of the OMPS instrument. The Ozone Processing Team has carefully
maintained the calibration of the nadir instruments through both hard and soft calibration
techniques.





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



**Figure Captions**

Figure 1. Each orbit of NM data measures a swath of total column ozone. 35 individual ozone measurements (see example near equator) are made for each scan line.

Figure 2. OMPS ozone will be compared with MOD (merged ozone data) ozone created by merging data from recent SBUV/2 instruments. Monthly average ozone for 60°S-60°N is plotted.

Figure 3. A comparison of OMPS NM ozone with average ozone from an ensemble of 52 northern hemisphere Dobson and Brewer stations, along with a linear fit to the data are shown. Weekly mean percent difference of OMPS NM ozone minus ground-based ozone is plotted.

Figure 4. For average ozone in the 60°S - 60°N latitude zone (lower panel), the average bias of NM ozone relative to MOD (upper panel) was reduced from 0.99% in version 1 to -0.20% in the version 2 processing.

Figure 5. A similar plot for the OMPS nadir profiler shows that the large bias in the released vsn 1 data is reduced in the vsn 2 processing.

Figure 6. In version 2 the four year average of March ozone latitude dependence (2013-2016) is shown in the lower panel for the mapper (dashed blue curve) and for the profiler (solid red curve). Percent differences from MOD are shown in the upper panel.

Figure 7. An average of ozone sonde data from Hilo Hawaii is compared with OMPS NP vsn 2 ozone profiles for coincident days, with percent difference plotted in the right panel.

Figure 8. The NP ozone anomaly, the difference from NOAA 19 ozone, for mid and low latitudes is shown as a function of time for total column ozone, the lower stratosphere, and the upper stratosphere. Ozone from the version 1 processing (in red) and the version 2 processing (in green) are shown.

Figure 9. OMPS NP v2 June zonal average ozone profiles (2012-2016) compared with NOAA 19 SBUV/2 profiles, MLS profiles, and profiles from the OMPS LP. OMPS NP vsn 2 percent differences from N19, MLS, and LP are plotted on the right.

Figure 10. The time dependence of the v2.0 ozone anomaly relative to NOAA 19 shown for low to mid latitudes.

Figure 11. The time dependence of the v2.0 ozone anomaly relative to NOAA 19 shown for high latitudes.



Figure 12. The time series of tropospheric ozone shown for four locations. Tropospheric ozone derived by subtracting OMPS LP stratospheric ozone from NM total column ozone is shown in the blue solid curve, while tropospheric ozone derived by subtracting MLS stratospheric ozone from OMI total column ozone is shown in the dashed red curve.



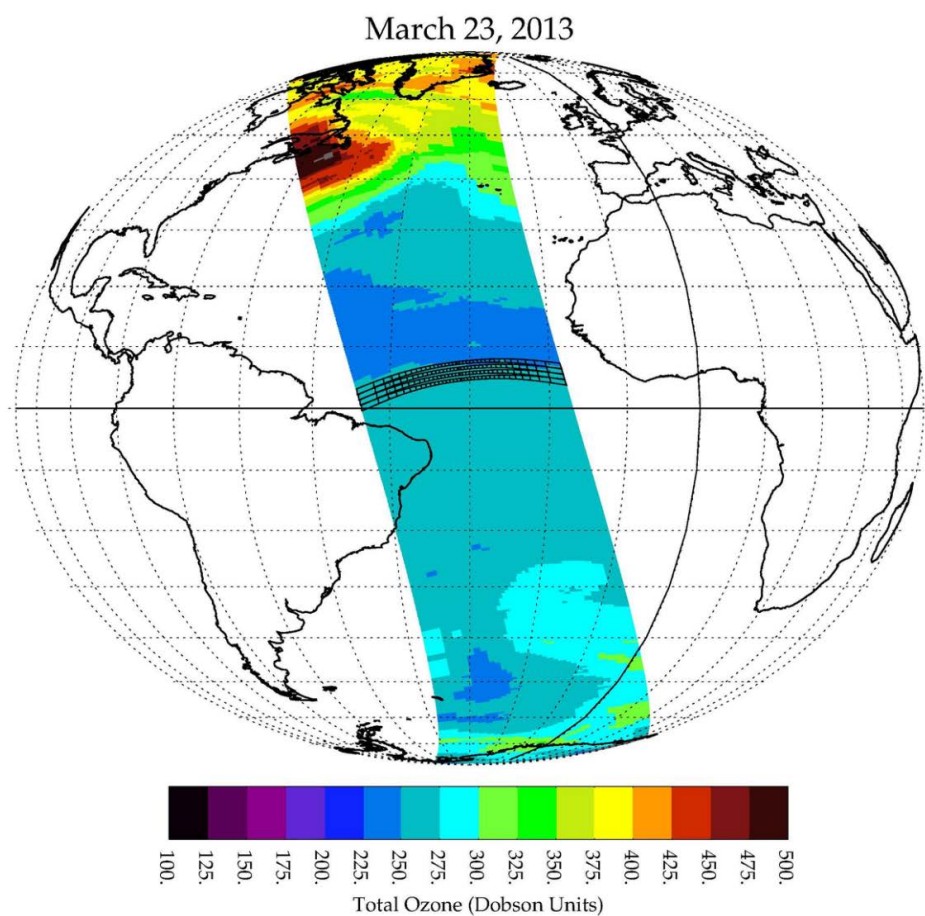

469     Figure 1



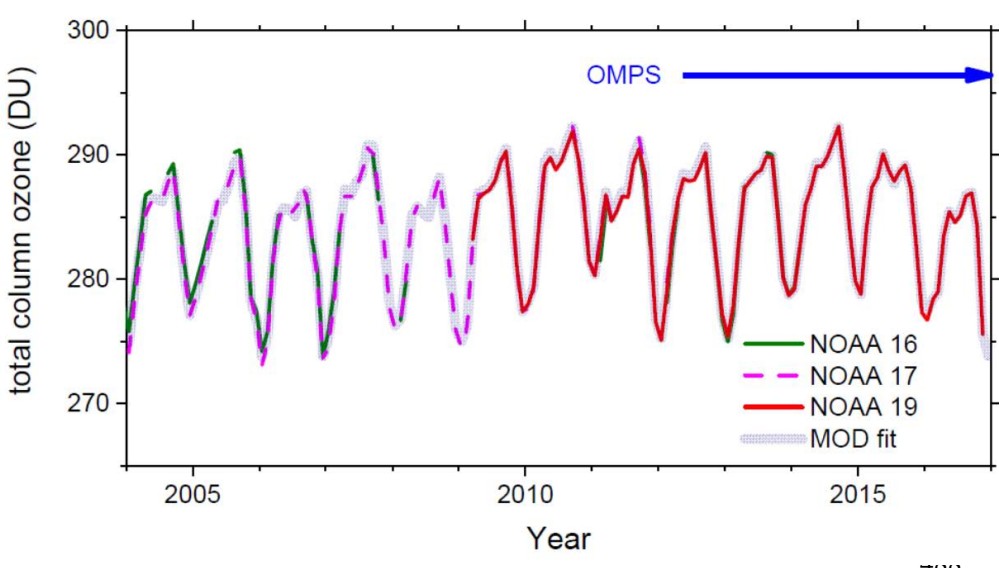

Figure 2





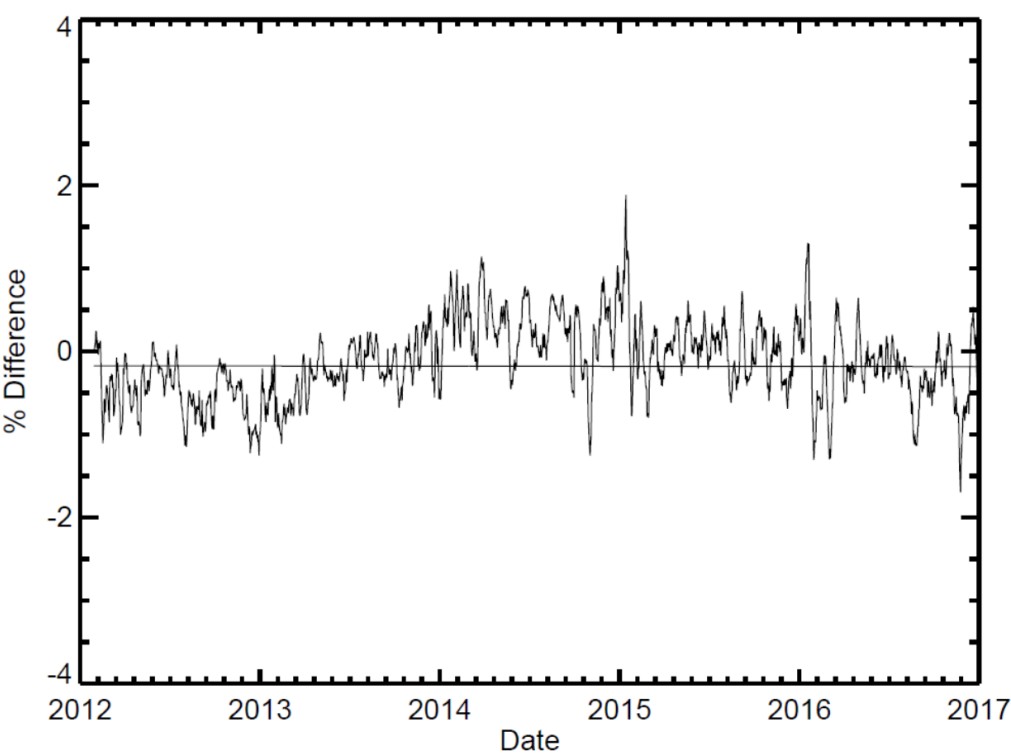

Figure 3





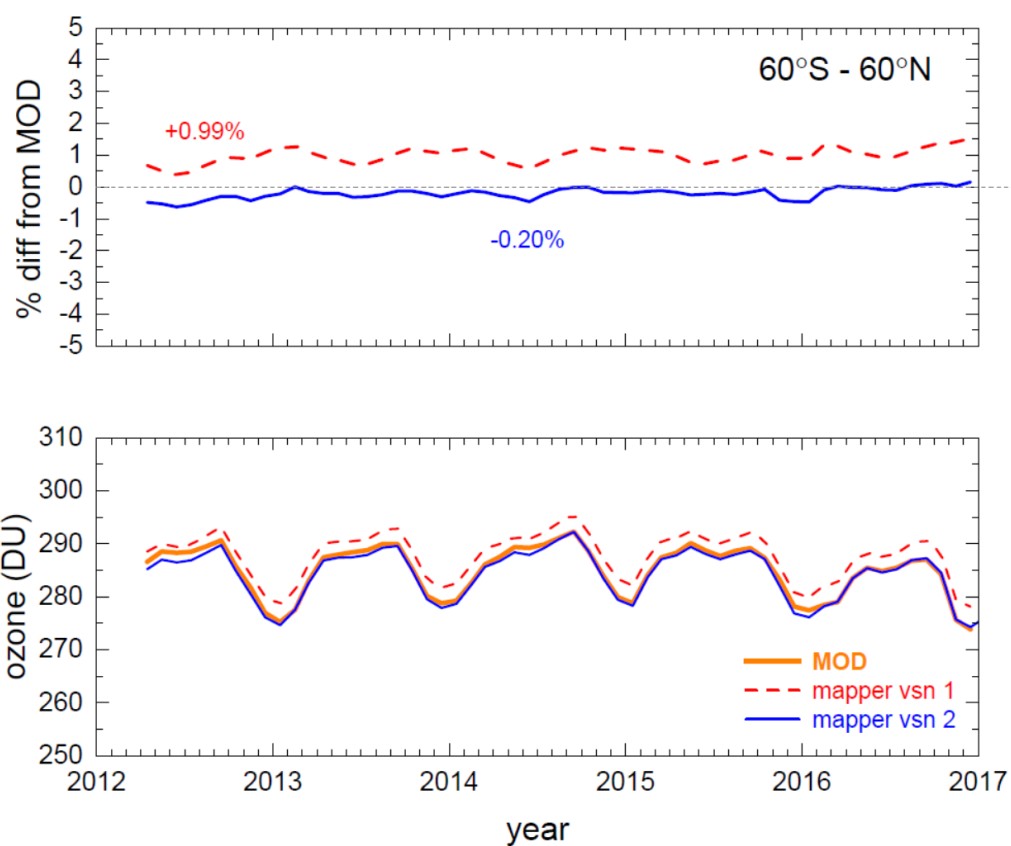

Figure 4





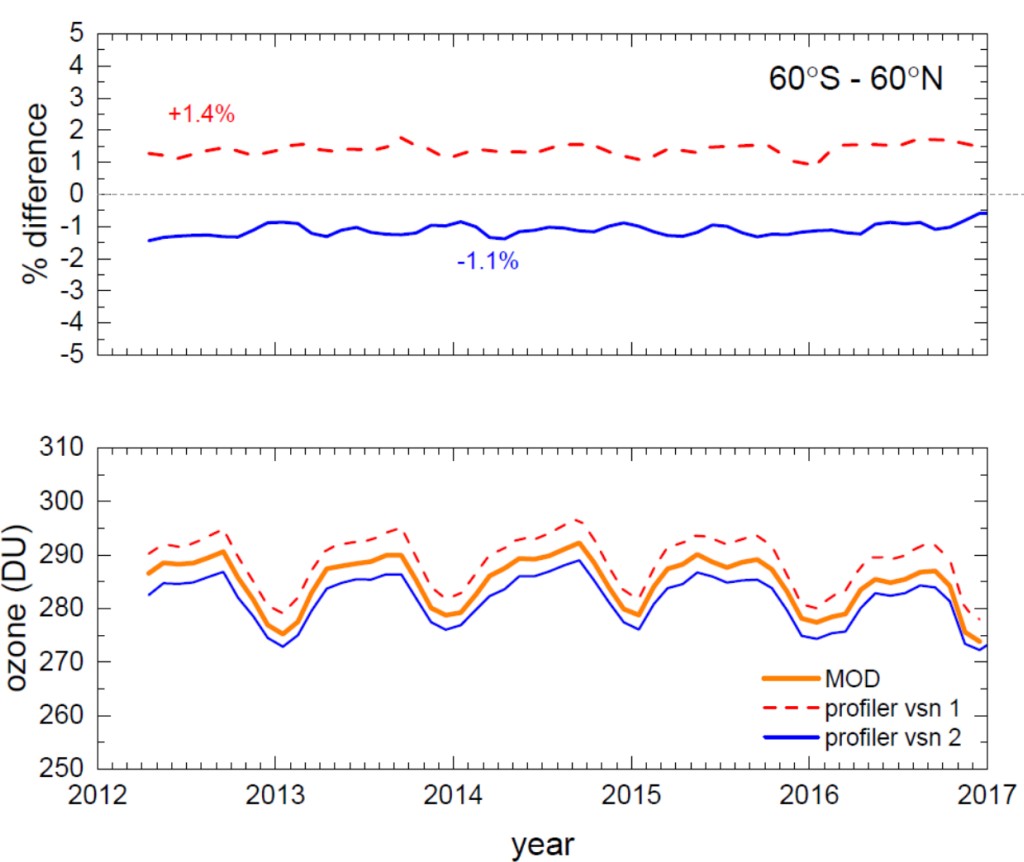

Figure 5


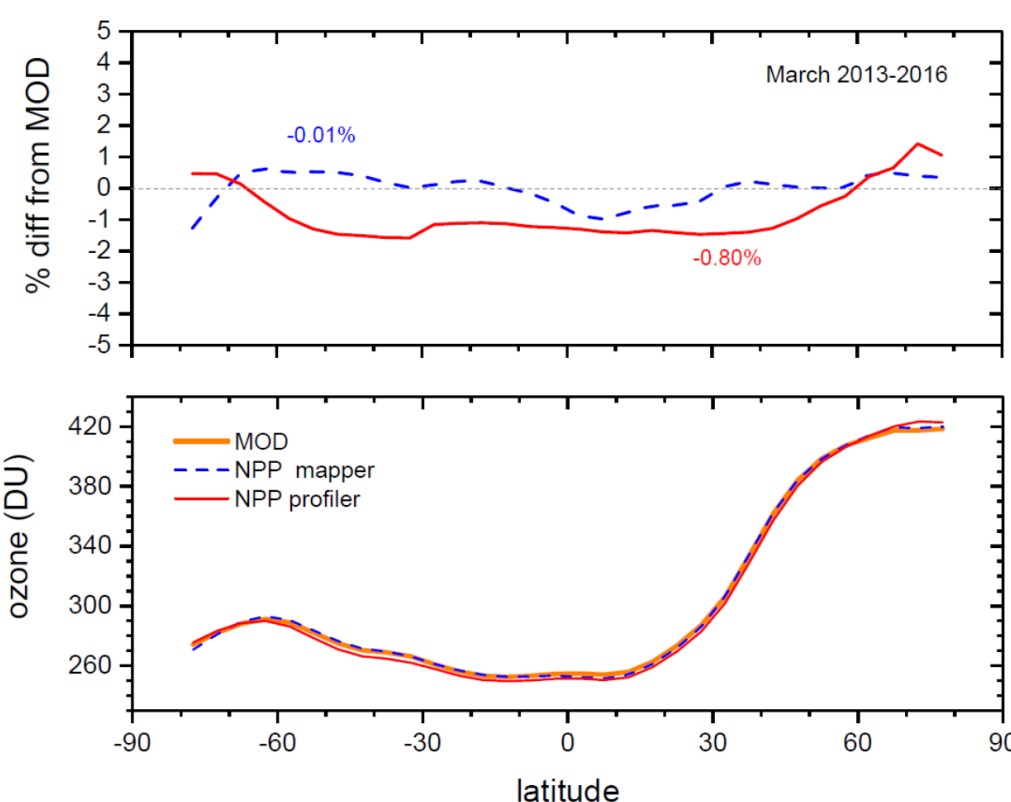

Figure 6



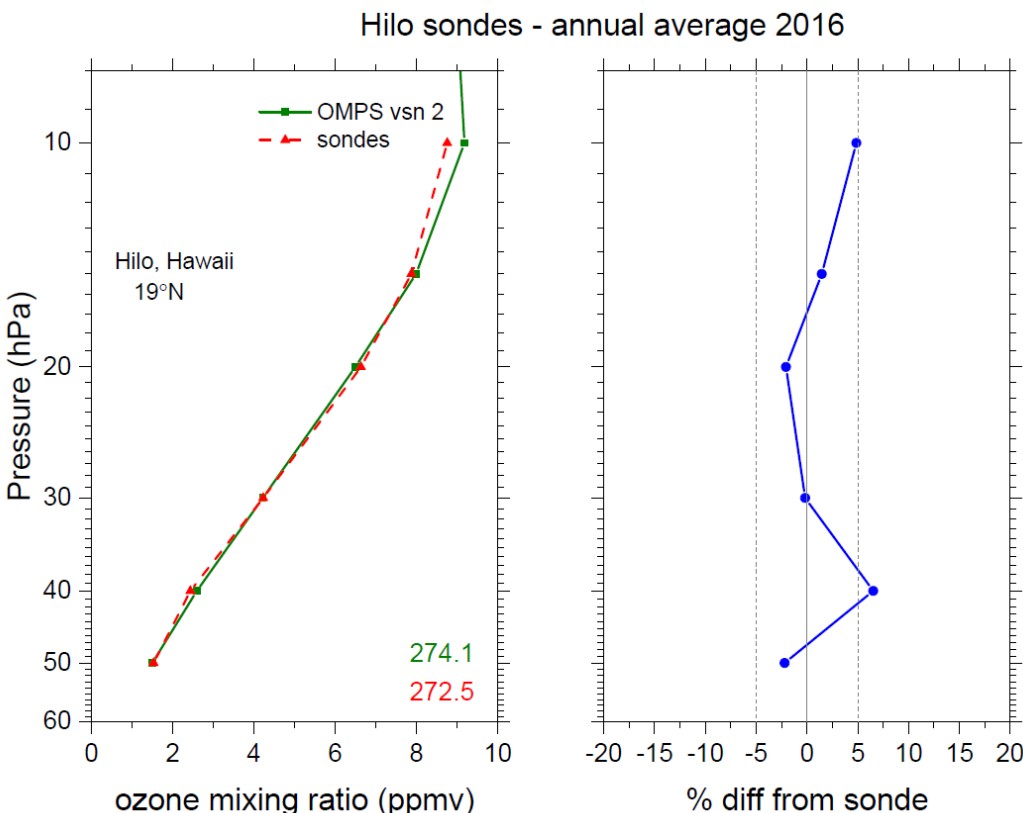

Figure 7



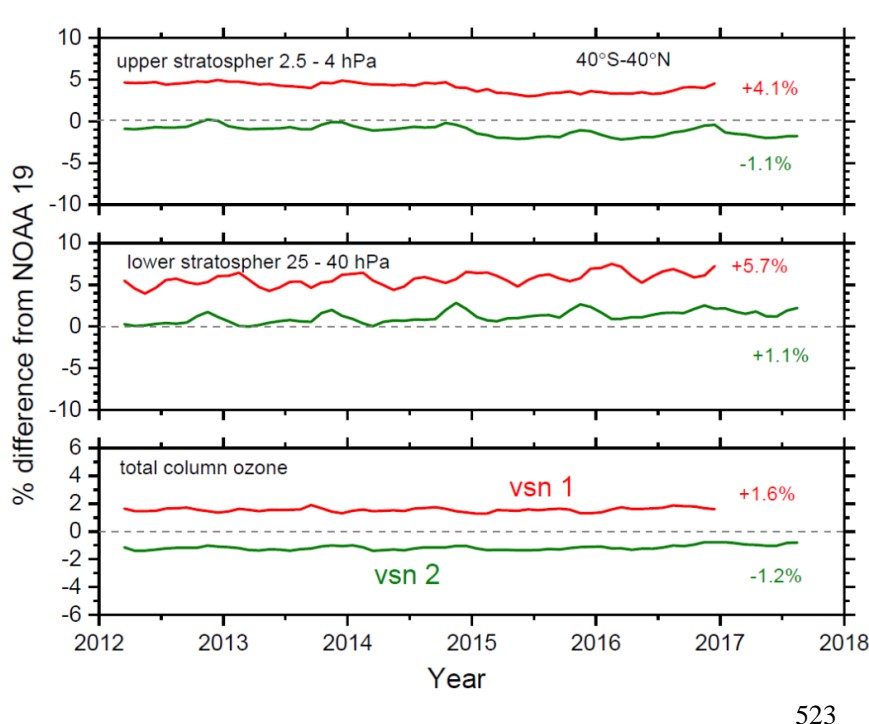

523

524        Figure 8



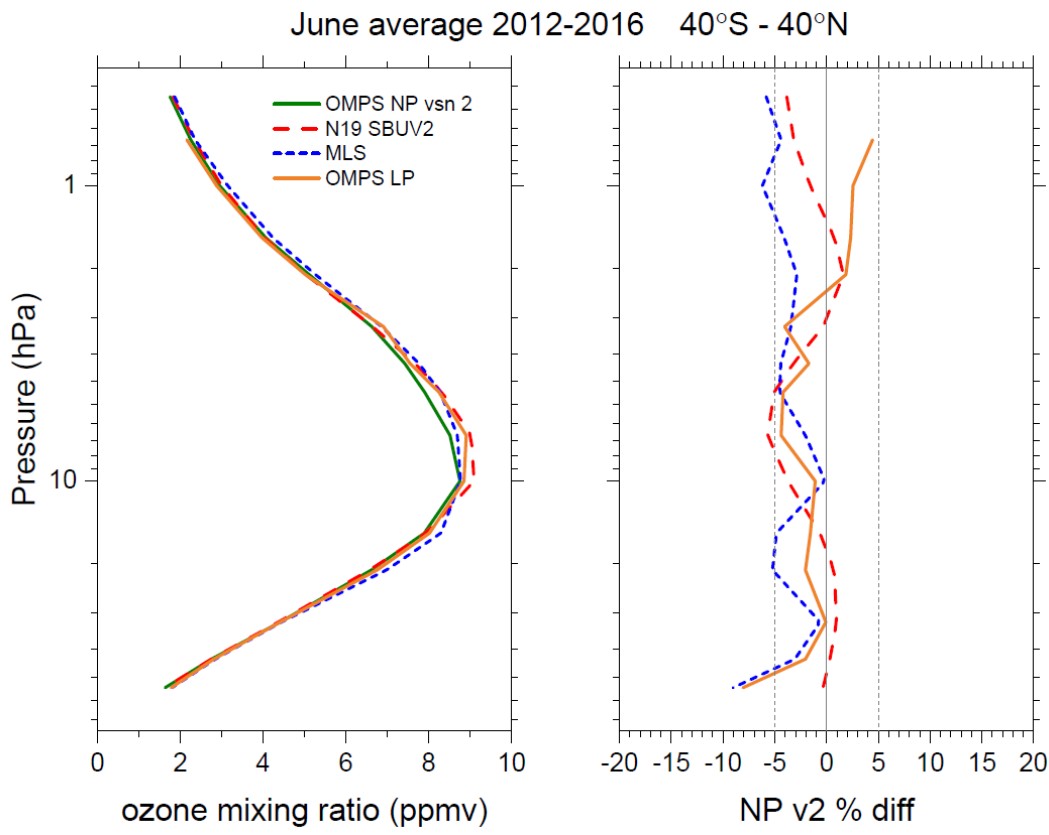

525
526      Figure 9





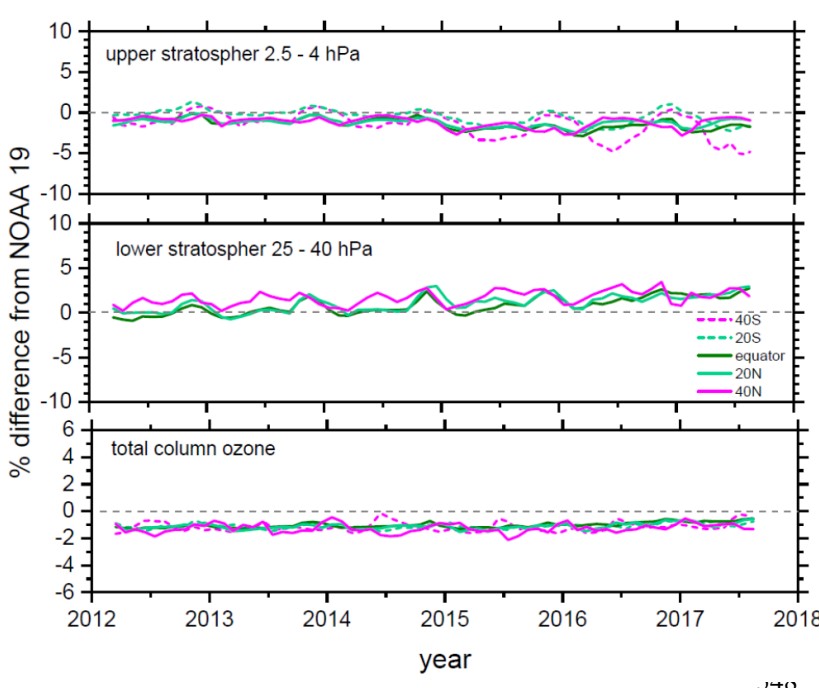

Figure 10





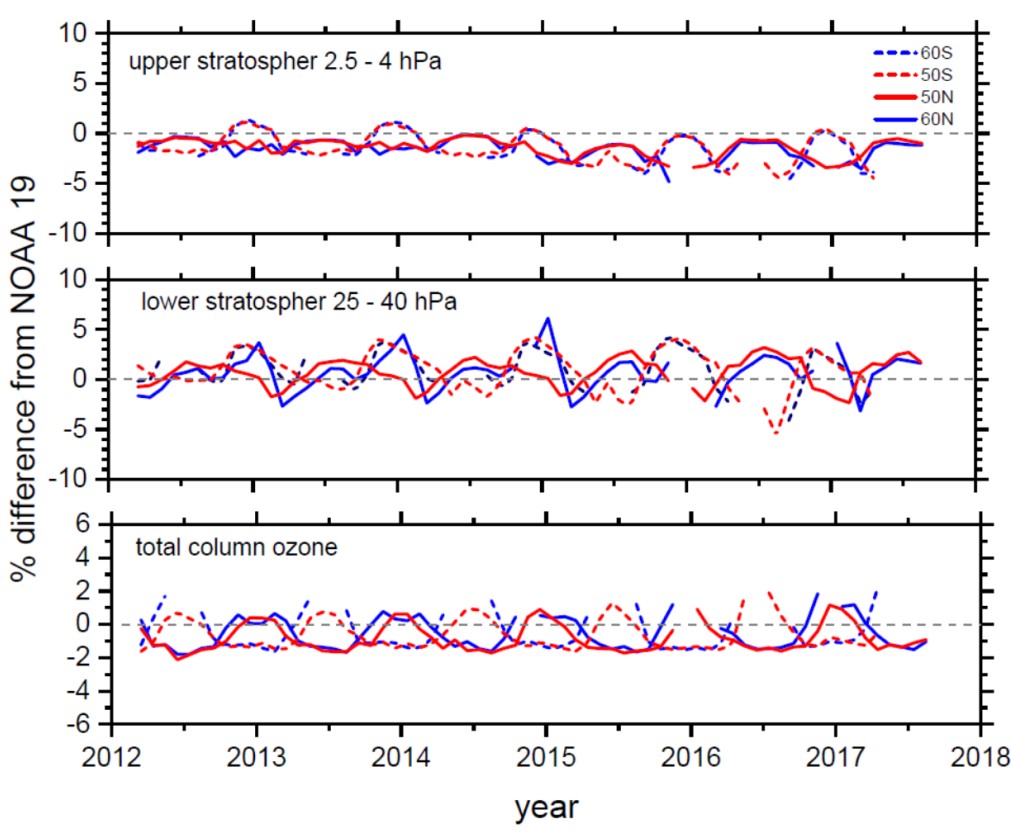

550       Figure 11




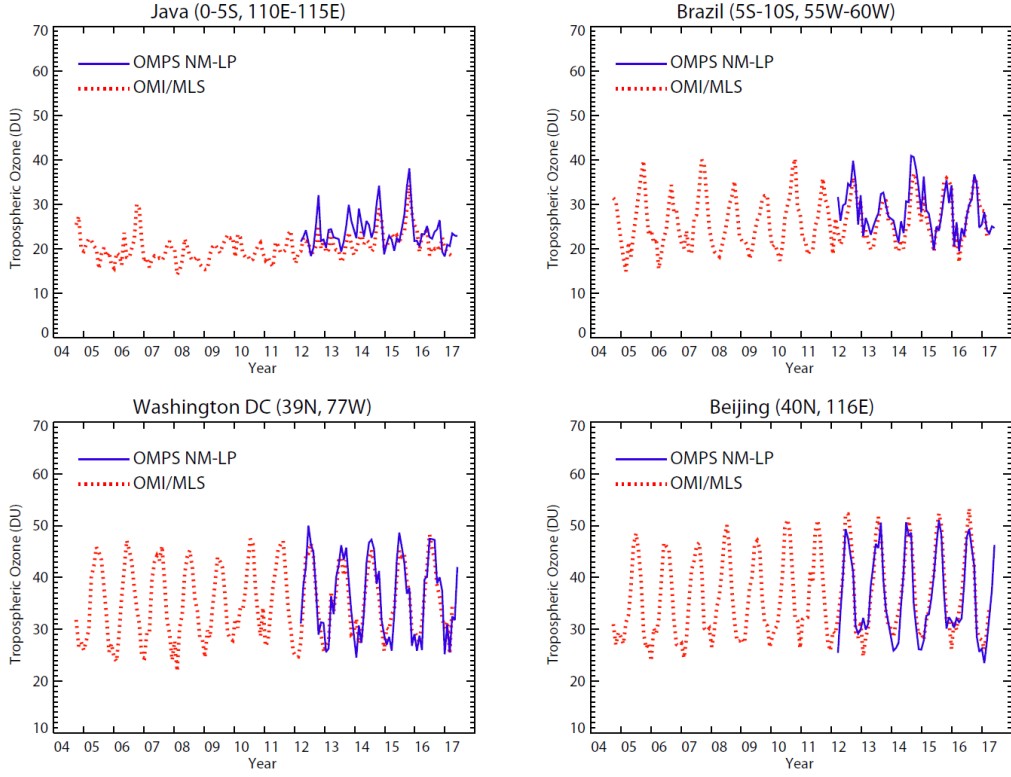

Figure 12