# Peer review of "Trend Quality Ozone from NPP OMPS: the Version 2 Processing"

_Atmospheric Measurement Techniques, 2018_

## Referee Comment (RC1) · Anonymous Referee #1 · 9 Aug 2018

This study provides interesting information on the new version of the NPP OMPS nadir total and profile data sets v2. The improvement is appropriately illustrated with a number of comparisons with independent measurements. The topic fits well in the field of interest of AMT. I have a number of recommendations that are worth to be considered before final publication.

**Major comments:**

- Soft calibration techniques are applied as a function of the cross-track to improve the measured reflectivity. Please mention if this is part of the new L1 version or if this soft-calibration step is applied as part of the ozone retrievals. Could you add figures showing the soft-calibration factors and also the impact on total ozone as

a function of the row?

- Generally, the comparisons are carried out using data until end of 2016. Please extend the time series up to now. This is especially important for drift analysis.

- Validation of total ozone columns from the nadir mapper (figure 3): There are systematic differences between Brewer and Dobson observations. I don't think it is appropriate to mix together those two networks for validating satellite data. Could you redo Figure 3 for each type of ground-based instrument separately? Please add as well V1 on those plots. Could you also give the numbers resulting from the drift analysis, even if they're small or not significant? For complete documentation of the product quality, it would be good to show also satellite/Ground-based differences as a function of other key parameters such as SZA, cross-track index, . . .

- Figure 6: Please mention what is the upper SZA limit for which retrievals are performed. A similar plot for the ozone hole season would be welcome.

- For the profile comparisons, as it is stated, the vertical resolution of the NP retrievals is quite coarse, making a direct comparison with sondes or limb profiles not necessarily straightforward. Did you apply averaging kernels to do the comparisons? Could you give more details on how this has been dealt with.

- Lines 245-247: In which conditions are the longer wavelengths from the NM used? Is it at higher SZA/latitudes only? In that case, the low bias cannot be explained by the use of those wavelengths? Could you clarify this please?

- Figures 10/11: it seems that there is a drift in the upper stratosphere, which should be mentioned even its amplitude is not that large.

- Tropospheric Ozone: Can tropospheric columns be derived by combining the NM and NP retrievals (instead of using the LP product)? How would such columns

compare with the presented product? Or are there good reasons to prefer using LP measurements to remove the stratospheric part? Is it possible to add a comparison with integrated tropospheric columns from sondes?

**Minor comments:**

- Lines 181-183: please rephrase as you state at lines 195-197 that the drift likely originates from the NOAA 19 calibration.

- Figure 7: add longitude of Hawaii, add comparison with version 1 as well if possible and state the meaning of the two numbers in green and red.

- Please, add also the new L1 version and availability in section 7.

- Line 314: "of" instead of "or"

- Lines 318-319: Mention again the possible link between the negative bias in total ozone from NP and the observed negative bias in the 6-10 hPa region of the profiles.

- Figure 2: there is a half-cropped number at the bottom-right of the figure

[Figure]

---

## Referee Comment (RC2) · Anonymous Referee #2 · 27 Aug 2018

The paper "Trend Quality Ozone from NPP OMPS: the Version 2 Processing" by McPeters et al. describes the latest version (2) of the OMPS Ozone column retrieval and a comparison to existing data sets mainly the SBUV/2 on NOAA 19. The data agree well and the remaining differences are small and cause further investigations. Despite the small difference the data reached a sufficient quality for trend analysis.

The paper is well written and clearly structured in sections containing comparisons of total columns, nadir profiles and tropospheric data. However the paper's title "version 2 processing" indicates that the algorithm is described in detail, but this section is rather short. The basic algorithms have been described in previous publication and it is well referenced here. Despite the reference a short summary of the algorithms should be given here. If I understand the paper correct the key improvement to the version 1

processing is the soft calibration that is applied now. The description of the calibration algorithm is short and some more details can be added.

Minor comments:

The AMT guidelines recommend adding a short summary of the paper and the sections at the end of the introduction. The authors may extend the last section of the introduction for this purpose.

Line 83: "An algorithm uses . . . ", add a reference to section 3.

Line 84: ". . . OMPS NM makes 400 individual scans per orbit with 35. . ." please add the resulting resolution in both dimensions (20000km / 400 scans $\sim$ 50 km)

Line 132: ". . .so NOAA 19 comparisons can be used for validation." This validation is not shown, please add.

Lines 222ff: "Looking at ground based comparisons of ozone in the lower stratosphere first, . . ." Are sondes really "ground based" measurements?

Line 244 f: "There is no evidence of a significant time dependent difference." I am sure the authors did some linear fits to the data and found the trend to be insignificant, however when looking at figure 8 it seems there is small decrease in the upper stratosphere and an increase in the lower stratosphere. So this statement might be clarified.

Line 250 f: "Selecting a single month for this comparison allows us to see any seasonal effect . . ." this sounds strange, a seasonal effect can only be seen if you use several months (minimum 4), and compare them to each other.

Line 294: The paper focuses on total column nadir profile so this section on tropospheric column can be seen as an "attachment" to illustrate the potential power of the dataset. Nevertheless the data might be compared to sonde data as well. The selected places for the comparison of the tropospheric columns are sounding stations for ozone sondes.

Line 295 "ozone from the NP plus LP combination" is it really the nadir profiler (NP) that is meant here not the nadir mapper (NM) as mentioned in line 287.

Line 316 "the drifting orbit" shall be mentioned earlier in the paper as a possible cause for the observed inconsistency.

Figure 3: Throughout the paper NOAA 19 is used as the key reference, for various comparisons. The validation of NOAA 19 is of course not subject of this paper, but nevertheless it might be nice to have the NOAA 19 data included in the comparison to the ground based observation.

---

## Author Comment (AC1) · 31 Aug 2018

"Soft calibration techniques are applied as a function of the cross-track to improve the measured reflectivity. Please mention if this is part of the new L1 version or if this soft-calibration step is applied as part of the ozone retrievals. Could you add figures showing the soft-calibration factors and also the impact on total ozone as a function of the row?"

The cross track corrections are applied as N value corrections in the algorithm. The corrections are needed for all wavelengths, not just the reflectivity wavelengths, so a wavelength dependent correction is applied. This is an important correction, but I feel that plots showing this would be overkill for a validation paper.

[Figure]

"Generally, the comparisons are carried out using data until end of 2016. Please extend the time series up to now. This is especially important for drift analysis."

The comparisons for the version 2 data have been extended through 2017. (See Figures 4, 5,8,10, and 11) Version 1 data were only processed through the end of 2016.

"Validation of total ozone columns from the nadir mapper (figure 3): There are systematic differences between Brewer and Dobson observations. I don't think it is appropriate to mix together those two networks for validating satellite data. Could you redo Figure 3 for each type of ground-based instrument separately?"

There have been many papers comparing Dobson and Brewer, and while there are some differences, we feel that there is a statistical advantage in using the aggregated data. As Balis et al and Van Roozendael et al. have noted, Brewer and Dobson "can agree within 1% when major sources of discrepancy are accounted for."

"Figure 6: Please mention what is the upper SZA limit for which retrievals are performed. A similar plot for the ozone hole season would be welcome."

The NM retrievals go up to 84 degrees SZA (results are flagged between 84 and 88). The profile algorithm has increased sensitivity to upper stratospheric ozone at high solar zenith angles so profiles are retrieved up tp 88 degrees. The same plot for September is not significantly different except that both NM and NP have a +1% offset at 70S, near the edge of the ozone hole. I don't think it adds much.

"For the profile comparisons, as it is stated, the vertical resolution of the NP retrievals is quite coarse, making a direct comparison with sondes or limb profiles not necessarily straightforward. Did you apply averaging kernels to do the comparisons? Could you give more details on how this has been dealt with."

For some purposes such as comparison of relative sensitivity for measuring the QBO, it is essential to apply the SBUV averaging kernel to the higher resolution data set (MLS in our recent study). This not an issue for NP vs NOAA 19 SBUV/2 of course since

they have the same resolution. The comparison with MLS shown in Figure 9 is a broad average (40S to 40N) and a 5 year average of June data. The MLS comparison actually shows less vertical structure than the N19 comparison. Using averaging kernels would make very little difference here.

"Lines 245-247: In which conditions are the longer wavelengths from the NM used? Is it at higher SZA/latitudes only? In that case, the low bias cannot be explained by the use of those wavelengths? Could you clarify this please?"

The longest wavelength used varies from 302 nm at small solar zenith angles (SZAs) to 317.5 nm at large SZAs. This was done to minimize the effect of smoke and mineral dust aerosols that have very high absorption in the UV. You are correct that this would have little effect on the bias so the text has been changed to reflect this. As we have studied this issue since the first draft of this paper was written, the use of the NM wavelengths in the NP retrieval looks less likely to be the source of the inconsistency.

"Figures 10/11: it seems that there is a drift in the upper stratosphere, which should be mentioned even its amplitude is not that large."

You are correct, and the drift is even more clear when the data are extended through the end of 2017. As stated in the paper, we suspect that the drift is mostly due to the aging N19 instrument.

"Tropospheric Ozone: Can tropospheric columns be derived by combining the NM and NP retrievals (instead of using the LP product)? How would such columns compare with the presented product? Or are there good reasons to prefer using LP measurements to remove the stratospheric part? Is it possible to add a comparison with integrated tropospheric columns from sondes?"

No, combining NM and NP does not add information on the troposphere. NP simply does not have the vertical resolution that would be needed. In principle the total ozone from NP should be more accurate than that from NM, but one of the conclusions of this

paper is that there are calibration issues with the NP that we are trying to resolve. The value of using stratospheric ozone from LP is that the measure of stratospheric ozone is much more accurate than that from NP.

Corrections were made based on the minor comments.

---

## Author Comment (AC2) · 31 Aug 2018

Despite the reference a short summary of the algorithms should be given here.

The basics of the algorithm are already described at what I think is the right level of detail in the last two paragraphs of section 3. The references are there if the reader needs real detail.

The AMT guidelines recommend adding a short summary of the paper and the sections at the end of the introduction. The authors may extend the last section of the introduction for this purpose.

This was done separately, not as part of the paper itself.

[Figure]

Line 83: "An algorithm uses ... ", add a reference to section 3.

Reference added.

Line 84: "... OMPS NM makes 400 individual scans per orbit with 35...:" please add the resulting resolution in both dimensions (20000km / 400 scans 50 km)

This information added.

Line 132: "...so NOAA 19 comparisons can be used for validation." This validation is not shown, please add.

Not sure what you mean. The MOD data used for validation in Figures 4, 5, and 6 is mostly based on NOAA 19 SBUV in the later years, and in Figures 8, 10, and 11 the validation comparisons are explicitly against NOAA 19. A comparison of NOAA 19 ozone to the Brewer/Dobson network has been added.

Lines 222: "Looking at ground based comparisons of ozone in the lower stratosphere first, ..." Are sondes really "ground based" measurements?

Since each sonde is prepared and launched from a single ground site, I would consider it ground based. Could be a small confusion in terminology I guess.

Line 244: "There is no evidence of a significant time dependent difference." I am sure the authors did some linear fits to the data and found the trend to be insignificant, however when looking at figure 8 it seems there is small decrease in the upper stratosphere and an increase in the lower stratosphere. So this statement might be clarified.

Adding all the data for 2017 makes the middle stratosphere time dependence even more clear, so a comment on this has been added to the text.

Line 250: "Selecting a single month for this comparison allows us to see any seasonal effect ..." this sounds strange, a seasonal effect can only be seen if you use several months (minimum 4), and compare them to each other.

[Figure]

The sentence is saying that an annual average would hide small seasonal variations.

Line 294: The paper focuses on total column nadir profile so this section on tropospheric column can be seen as an "attachment" to illustrate the potential power of the dataset. Nevertheless the data might be compared to sonde data as well. The selected places for the comparison of the tropospheric columns are sounding stations for ozone sondes.

The ability to derive accurate tropospheric ozone by subtracting stratospheric ozone depends on high accuracy of the original total column ozone product, which is the subject of this paper. So yes, the purpose of this section is to illustrate the "potential power" of this dataset. We are working on implementing a tropospheric ozone product, and papers detailing the results will be forthcoming.

Line 295: "ozone from the NP plus LP combination" is it really the nadir profiler (NP) that is meant here not the nadir mapper (NM) as mentioned in line 287.

No, we are exploring the combination of the high resolution limb profiler (LP) and the low resolution nadir profiler (NP). Since the two have different strengths and weaknesses, the combination might be more accurate - to help reduce limb pointing errors for instance, or to refine the nadir a priori profile.

Line 316: "the drifting orbit" shall be mentioned earlier in the paper as a possible cause for the observed inconsistency.

Good suggestion. Added comment to the discussion of the time dependence seen in Figure 4.

Figure 3: Throughout the paper NOAA 19 is used as the key reference, for various comparisons. The validation of NOAA 19 is of course not subject of this paper, but nevertheless it might be nice to have the NOAA 19 data included in the comparison to the ground based observation.

Good suggestion. Have added a N19 comparison with the Brewer/Dobson network to

Figure 3.

[Figure]